# Chlorine Dioxide Degradation Issues on Metal and Plastic Water Pipes Tested in Parallel in a Semi-Closed System

**DOI:** 10.3390/ijerph16224582

**Published:** 2019-11-19

**Authors:** Alberto Vertova, Alessandro Miani, Giordano Lesma, Sandra Rondinini, Alessandro Minguzzi, Luigi Falciola, Marco Aldo Ortenzi

**Affiliations:** 1Department of Chemistry, Università degli Studi di Milano, Via Golgi 19-20133 Milan, Italy; alberto.vertova@unimi.it (A.V.); giordano.lesma@unimi.it (G.L.); sandra.rondinini@unimi.it (S.R.); Alessandro.Minguzzi@unimi.it (A.M.); luigi.falciola@unimi.it (L.F.); 2Department of Environmental Science and Policy (ESP), Università degli Studi di Milano, Via Celoria 2-20133 Milan, Italy; alessandro.miani@unimi.it; 3Italian Society of Environmental Medicine (SIMA), Via Monte Leone 2-20149 Milan, Italy; 4CRC Materiali Polimerici (LaMPo), Department of Chemistry, Università degli Studi di Milano, Via Golgi 19-20133 Milan, Italy

**Keywords:** chlorine dioxide, water disinfection, plastic pipes, metal pipes, microplastics, water treatment, degradation by-products

## Abstract

Chlorine dioxide (ClO_2_) has been widely used as a disinfectant in drinking water in the past but its effects on water pipes have not been investigated deeply, mainly due to the difficult experimental set-up required to simulate real-life water pipe conditions. In the present paper, four different kinds of water pipes, two based on plastics, namely random polypropylene (PPR) and polyethylene of raised temperature (PERT/aluminum multilayer), and two made of metals, i.e., copper and galvanized steel, were put in a semi-closed system where ClO_2_ was dosed continuously. The semi-closed system allowed for the simulation of real ClO_2_ concentrations in common water distribution systems and to simulate the presence of pipes made with different materials from the source of water to the tap. Results show that ClO_2_ has a deep effect on all the materials tested (plastics and metals) and that severe damage occurs due to its strong oxidizing power in terms of surface chemical modification of metals and progressive cracking of plastics. These phenomena could in turn become an issue for the health and safety of drinking water due to progressive leakage of degraded products in the water.

## 1. Introduction

In the last decades, there has been a growing interest in the use of chlorine dioxide (ClO_2_) as an efficient drinking water disinfectant [1,2,3,4,5] versus other disinfectants, such as chlorine-free (typically deriving from either sodium hypochlorite, calcium hypochlorite or Cl_2_) or monochloramine because its strong oxidizing power is capable of eliminating viruses and chlorine-resistant pathogens (for example in Legionella surveillance [6]), as well as preventing biofilm formation. 

Moreover, one of the advantages of ClO_2_ is that it does not lead to the production of trihalomethanes (THMs), some of which (e.g., chloroform) are carcinogens. For this reason, disinfection with ClO_2_ is frequently used in water that is particularly prone to THM formation [7].

Recommended doses may differ according to the way ClO_2_ is produced and put into the potabilization process, and according to the quality of the water and to legislative limits. If used as a residual disinfectant, the maximum concentration is 0.8 mg/L ClO_2_ in the USA [8] and 0.2 mg/L ClO_2_ in Germany [9], while in other countries like Canada and Italy no upper limits are set. In most cases, maximum concentration limits are referred to the water coming out from the tap, therefore the quantity of disinfectant introduced upstream is higher: only in a few cases do defined rules exist about the limits of ClO_2_ concentration at the beginning of the water stream, such as in Germany where a maximum dosage of 0.4 mg/L of ClO_2_ is allowed [9] and in the U.K., which sets a limit of 0.5 mg/L as the sum of ClO_2_, chlorites, and chlorates [10].

Beside its use as a secondary disinfectant in potable water treatment, when it is used in sanitary hot water recirculation loops in complex systems, such as hospitals, a typical dosage is done in order to achieve 0.2–0.3 mg/L ClO_2_ on the water coming out of the tap [6].

Notwithstanding these important properties, its strong oxidizing power makes chlorine dioxide very aggressive towards the materials conventionally used to produce water pipes, i.e., plastics and metals.

The most commonly used plastic materials for the production of pipes belong to the polyolefins family. Polyethylene (PE) and polypropylene (PP) have been widely used in the drinking water distribution network and households’ installation both for multilayer pipes and as self-standing materials. However, practical experience has demonstrated that a significant number of PE and PP pipes fail prematurely when exposed to drinking water containing ClO_2_ [11,12,13,14,15,16,17].

To prevent degradation during processing and to extend product service life, antioxidants are incorporated into polyolefins. Models have been developed that predict antioxidant loss by migration to the surrounding media. However chlorine dioxide, which is known as an energetic oxidant capable of rapidly oxidizing phenolic compounds, degrades antioxidants very rapidly and, when the antioxidant system has become totally depleted, a fast but strictly surface-confined degradation of polyolefins occurs [14,15,18,19]. Moreover, Bredacs et al. believe that degradation is due to ClO_2_ attacking simultaneously both antioxidants and the polymers [20].

From a general point of view, the macroscopic mechanism responsible for final pipe degradation when used with water containing any of the three disinfectants chlorine conventionally employed (free chlorine, chloramines, or chlorine dioxide) is considered to be the same, i.e., depletion of stabilizer at the inner pipe surface, oxidation of the inner layer due to breaking of the carbon–hydrogen or carbon–carbon bonds, microcracking of the inner layer due to chemi-crystallization [21], crack propagation through the wall with oxidation in advance of the crack front, reduction of molecular weight which decreases the tensile strength of the polymer, and final rupture of the remaining pipe, [22] ultimately resulting in pipe failure.

Many scientists have observed that chlorine dioxide is more aggressive than other disinfectants against polyolefins (polyethylene, polypropylene, and polybutylene). One explanation for this could be the fact that chlorine dioxide is a dissolved gas, which diffuses into the polymer more readily than other disinfectants. In addition, as stated above, chlorine dioxide heavily reacts with phenols. This is one of the advantages of chlorine dioxide as a disinfectant, but since the long-term stabilizers are usually hindered phenols, this will lead to a rapid reaction with the stabilizer, making the material susceptible to oxidative degradation.

In the case of metals, some studies exist on the detrimental effect of chlorine dioxide on the inner surface of pipes. Vidic et al. [23] studied this effect in two of the most common water metal pipe materials, i.e., copper and galvanized iron. Using distilled water containing 1 mg/L ClO_2_, they found out that ClO_2_ on one side significantly contributes to the corrosion of metals and, on the other side, it is consumed due to the corrosion process, with Fe_3_O_4_ and Cu_2_O as the main degradation products of galvanized steel and copper pipes respectively, acting as consumers of ClO_2_. Other studies [24] confirm that, in the case of copper, oxides are the main products of ClO_2_ degradation and that the formation of byproducts should be carefully considered in drinking water distribution systems containing copper pipes.

Nevertheless, in the present scientific literature dealing with the effect of chlorine dioxide on commercial metal pipes, some gaps still exist. One is related to the focus of previous studies, which frequently deal with either the determination of ClO_2_ concentration decay due to the pipes [23,24] or with the overall quality of water treated with chlorine dioxide [6] and passed through pipes in a real, but confined (i.e., a single building), environment.

It is difficult, especially in the case of metal pipes, to find studies aimed at defining the possible degradation of metal pipes themselves due to water treated with ClO_2_, even if the issue is well known in the commercial field.

The other important issue, involving studies on plastic, multilayer (i.e., polymer/metal/polymer), and metal pipes in water-containing disinfectants, is related to the experimental setup used for testing water and pipes. It is extremely difficult to reproduce conditions simulating a real environment due to a number of different factors, some of them being, for example, the long times required to observe the degradation of pipes with low disinfectant concentrations or the need for a constant dosage in continuously flowing water. 

Gedde et al. focused on the creation of models to study of degradation of pipes using a simulant of polyolefins, such as squalane, to reduce the influence of additives or other factors [19,25].

In other cases, pipes are tested in a “closed” system containing high quantities of disinfectant in water, to study the quality of water and/or the quality of the pipes in relatively short aging times [26,27].

Last but not least, when pipes put in a real environment are studied, i.e., in hospitals [6] or in other buildings, it is very hard to trace the complete water transport system from the source of water to the last mile, and therefore results cannot give a reliable estimation of the quality of pipes subjected to the flow of water in the presence of disinfectants. 

In this context, in the present paper a comprehensive study is presented, using a test system resembling ASTM F-2023: new pipes, bought from the market, were joined and put in a “semi-open” system, where a drinkable water flux containing ClO_2_ was recirculated. Two plastic-based pipes were tested, namely, one based on random polypropylene (PPR) and the other based on multilayers made by polyethylene of raised temperature (PERT) and aluminum, along with two types of metal pipes made of copper and galvanized steel.

The pipes were tested for 8 weeks and then analyzed to assess the effects of ClO_2_ on the overall quality of the used materials.

## 2. Materials and Methods 

### 2.1. Experimental Setup

The accelerated aging test was performed using water at 70 °C, pumped with a pressure of 5 bar with a rate of the fluid of 0.5 m/s, pH = 7, and a concentration of chlorine dioxide of 1 mg/L ClO_2_. The raw water was produced using reverse osmosis filtration apparatus BWE 2084 supplied by Adenco Srl.

The chlorine dioxide target concentration was obtained by dosing a low pH commercial stabilized solution (nominal concentration of 1 g/L ClO_2_) supplied by Sanipur Srl. To achieve the target pH, a 0.01 M sodium hydroxide (NaOH) solution was dosed. ClO_2_ and NaOH were dosed continuously with electromagnetic diaphragm pumps controlled by a programmable logic controller (PLC).

The pH, conductivity, and ClO_2_ concentration were constantly monitored with a Memosens CPS31D (pH-meter with a saturated calomel reference electrode), Condumax CLS21D (digital conductivity sensor), and CCS240 Analog amperometric sensor, respectively, supplied by Endress & Houser.

A continuous purge of the test fluid was applied to prevent the accumulation of disinfection by-products. The purged test fluid was then conveyed through a filter containing active carbon and, after that, was sent to the reverse osmosis system. A fraction of this water was reused as a raw water source. 

The target temperature was maintained with a titanium-based heat exchanger continuously supplied with steam. The quantity of steam necessary for heating was regulated using a PLC.

All the pipes conveying the fluid test circulating in the pipe samples were made of titanium and properly insulated.

Pipe samples were positioned in a thermostatic chamber at the same temperature of the test fluid (70 °C).

Copper pipes were joined using brass compression fittings, galvanized steel pipes were joined with threading, PPR pipes were welded, and PE-RT pipes were joined with expansion fittings.

The apparatus was developed to perform the accelerated aging test of the whole system (pipes and joints) and to resemble a real-world scenario where, from the potabilization plant to the tap, the drinking water interacts with several kinds of materials. Tuning the purge flow rate, for example, the accumulation of disinfection and corrosion by-products can be minimized or maximized. The concentration of ClO_2_ (1 mg/L ClO_2_) is a compromise between the need of an accelerating process (high concentrations), the necessity to have a model fluid with a concentration of sanitizer not too far from a real-world application and checking the experimental set-up already adopted by other research groups [20,23,28].

A scheme of the experimental set-up used is presented in Figure 1.

### 2.2. Characterization

#### 2.2.1. Fourier Transform Infrared Spectroscopy (FT-IR)

FT-IR Spectrometer Spectrum 100 (PerkinElmer Italia SpA, Milan, Italy) with an attenuated total reflection (ATR) was used to assess the influence of ClO_2_ on the inner part of PPR and multilayer PE-RT pipes. Analyses were conducted on the samples “as is” both on the surface and on the bulk of the pipe, 300 microns under the surface. The samples were analyzed with 8 scans in the 4000–380 cm^−1^ range.

#### 2.2.2. Differential Scanning Calorimetry (DSC)

Differential scanning calorimetry (DSC) analyses were conducted under nitrogen flow using a Mettler Toledo DSC 820 (Mettler Toledo SpA, Novate Milanese (MI) – Italy), on PPR and PE-RT samples weighing from 4 to 6 mg each. Samples were placed in a 40 µl aluminum pan, taking care of having the maximum contact area possible between the sample and the pan. Samples first heated from 25 °C to 200 °C at 10 °C/ min, left 2 min at 200 °C to eliminate internal stresses, cooled from 200 °C to 25 °C at −10 °C/min, left 2 min at 25 °C and then a second thermal cycle identical to the previous one was used to evaluate the material behavior. In this case, analyses were conducted on the surface and on the bulk of the materials.

#### 2.2.3. Optical Microscopy

Leica EZ4W Optical Microscope (Leica Microsistems Srl, Buccinasco (MI), Italy) was used to observe the inner surface of metal pipes: images were acquired with 35X magnification.

#### 2.2.4. Scanning Electron Microscopy (SEM)

Samples were analyzed using a Jeol 5500 LV Scanning Electron Microscope (Jeol (Italia) SpA, Milan, Italy) with an IXRF Energy Dispersive Spectrometer (EDS). The samples were analyzed under vacuum (20–30 Pa), using a potential of 20 KeV and a diameter of the beam of 1 µm.

#### 2.2.5. Electrochemical Impedance Spectroscopy

Electrochemical impedance measurements (EIS or IS) were conducted on a multilayer pipe and a metal, namely PERT pipes and copper to characterize the samples in terms of double-layer capacitance (*C*_dl_) for both samples, and also electron transfer resistance (*R*_CT_) in the case of the metal sample. For Cu pipes, EIS was carried out with a Frequency Response Analyzer Solartron 1260, coupled with Electrochemical Interface Solartron 1287, both driven by ZPlot^®^ and Corrware^®^ provided by Scribner. In the case of PERT samples, an IS dry procedure was adopted by sealing the testing tube on one side, filling it with 72.2769 g of mercury, used as an internal electrode connected directly to the investigated polymer layer, and using a Pt wire dipped in Hg. Externally, exposed aluminum was used as the electrode connected to a copper wire. A 4 wires measurement was adopted, using only FRA 1260 (see Appendix A) between 100 Hz and 10 MHz, with an AC amplitude of 500 mV.

Cu pipes were cut longitudinally in two pieces long about 2 cm, whose edges and external surfaces were isolated with an insulating paint provided by BLASBERG-ETHONE (code BN 26.823.4), to leave only the inner surface of the tube exposed, 2.3 cm^2^ for Cu *t* = 0 and 3.15 cm^2^ for Cu *t* = 8. These samples have been characterized by EIS measurements, using a 3 electrode setup cell: Cu sample as the working electrode; Pt sheet as the counter electrode, and the saturated calomel electrode (SCE) as the reference. In these measurements, the Electrochemical Interface Solartron 1287 was coupled with FRA, to perform frequency scans while the inner tube surface was polarized at selected potentials. Before EIS measurements, cyclic voltammetry studies were carried out, in 0.5 M KNO_3_, to select the potential values for performing the EIS studies, polarizing the active surface area of the sample between −0.2 and +0.2 V (SCE) at 20 mv s^−1^ scan rate. Hence, EIS measurements on the Cu samples were carried out at 0, 0.05, and 0.1 V vs. SCE using 10 mV AC amplitude and a frequency range from 0.1 Hz to 1 MHz.

## 3. Results and Discussion

### 3.1. Microscope Analyses

As described in the Materials and Methods section, pipes were kept for 8 weeks under a continuous flow of water containing 1 mg/L ClO_2_. Optical microscope images taken on metal pipes show marked changes with time, comparing the new materials and the aged ones, at the end of the test (after 8 weeks of aging) (Figure 2).

In particular, the black color of Cu pipes indicates the formation of cupric oxide, whereas the galvanized steel shows marked signs of degradation and oxidation.

SEM analyses were performed on both plastic and metal pipes, as shown in Figure 3.

In all cases, SEM images demonstrate severe degradation of the inner surface of the pipes even if, as expected, the behavior changes significantly according to the material, as illustrated after.

EDS analyses, shown in the Supporting Information, show that in all cases chlorine dioxide yields to a marked change in the composition of the constituent material after its aging. Table 1 shows the main elements found before and after 8 weeks of aging together with the relative quantities detected.

Even if the EDS results must be considered as semi-quantitative, they clearly indicate that all pipes have undergone severe degradation: Copper has turned to copper oxide, which, according to Figure 3b, is very porous. Galvanized steel has undergone partial dezincification, with the surface showing the typical appearance of degraded galvanized steel.

Plastic-based pipes have different degradation profiles, with the formation of cracks that, in the case of PE-RT, follow the extrusion direction of the pipe with perpendicular cracks between them, in a brick wall-like pattern. In the case of PPR, cracks are different, i.e., more random. In both cases, materials are highly oxidized on the surface, as confirmed by EDS.

In all pipes, small quantities of chlorine are present, probably as a result of ClO_2_ interaction with metals or organic fractions of the pipes themselves. This result seems to confirm what has been previously observed in relation to the decay of ClO_2_ concentration in exhausted water due to the interaction with pipes [23,24].

Copper is present in all pipes, indicating that, when pipes made with this material are connected to other pipes, aging leads to copper leakage on the inner walls of the pipes. This can, in turn, accelerate aging due to possible catalysis of degradation phenomena [29,30].

It has to be considered that all the elements found on the surface of aged pipes are likely to be transported over time along the pipelines up to the tap.

### 3.2. FT-IR and DSC Analyses on Plastic Pipes 

#### 3.2.1. FT-IR Analyses

FT-IR analyses were performed on PERT, on both the inner surface and the bulk (i.e., 300 microns under the surface) of the material after 4 and 8 weeks of aging, to detect any chemical modification of the polymers used.

Before aging, FT-IR spectra of PERT and PPR show weak absorption carbonyl signals in the 1750–1735 cm^−1^ region, attributable to the antioxidants present as additives in the polymer, which shift to lower values (1710–1715 cm^−1^) and increase in intensity, suggesting the formation of both carboxylic and carbonyl compounds in the highly oxidized samples (after 4 and 8 weeks of aging). Moreover, a broad hydroxyl band in the 3200–3500 cm^−1^ region appears, confirming the previous results (Figure 4, Figure 5, Figure 6 and Figure 7).

Furthermore, these bands are accompanied by the appearance of a broad absorption band between 1550–1620–1640 cm^−1^ that becomes dominant after 8 weeks and could be assigned to both H-bonded carboxyl and to carboxylate anion groups, the latter of which is derived from salification. All these data are consistent with the reported literature [15] and indicate a consistent degradation of the pipe inner surface caused by ClO_2_.

This phenomenon appears more moderate in the bulk of the pipes (i.e., 300 microns under the surface). In particular, in the case of PERT pipes, after 4 weeks of treatment with ClO_2_, only the depletion of the antioxidant can be observed by the disappearance of the week absorption around 1745 cm^−1^, while the long-term exposition (8 weeks) produced the formation of minimal amounts of carbonyl and carboxylic functional groups. Similar behavior was observed in the case of PPR pipes, except for a greater persistence of the antioxidants before the appearance of carbonyl and carboxylic functional groups in the IR spectra (Appendix A).

The mechanisms underlying polyolefin degradation, as seen by FT-IR, have been recently discussed, even if there is no agreement on the way ClO_2_ acts: some state that chlorine dioxide directly acts on the polymer whereas others believe that reduction products of ClO_2_ are responsible for plastic corrosion [15,16,31]. Van der Stok et al. [32] observed that in the case of PP, the presence of applied stress contributes to the occurrence of a rupture when chlorine dioxide is used, therefore oxidation and environmental stress-cracking are jointly responsible for the degradation of PP pipes exposed to ClO_2_. 

#### 3.2.2. DSC Analyses

DSC analyses were conducted to assess the thermal features of aged pipes, in terms of melting and crystallization temperatures and enthalpies, determining the changes in the crystallinity of the materials after prolonged exposure to ClO_2_. All DSC curves are given in SI.

Thermal parameters obtained via DSC are reported in Table 2, where T_c_ is the melt crystallization temperature (measured during the cooling phase), ΔH_c_ is the melt crystallization enthalpy, T_m_ is the melting temperature, and ΔH_m_ its enthalpy. χ_c_ is the crystalline fraction, calculated considering 286.7 J/g and 207 J/g as the crystallization enthalpy of 100% crystalline PE and PP, respectively.

Our thermal analyses were not focused on the issues related to oxidation induction time (OIT) and oxidation onset temperature (OOT), which have been deeply investigated in previous papers [15,16,17,19,20,25]: published results show that, in any case, OIT is severely affected by ClO_2_. 

In PERT, the DSC results show that the crystalline content of the material increases with aging time, passing from about 46.5% to 55.4%; interestingly, after both 4 and 8 weeks, the bulk of the material is more crystalline than the surface, even if the differences between surface and bulk become lower. Differences in ΔH_c_, which is always higher than ΔH_m_, are less marked, confirming that over time, PERT becomes more crystalline. 

No significant change is observed in relation to Tm, Tc, or to the shape of melting and crystallization peaks.

In the case of PPR, the polymer itself has a lower crystalline content than PERT and a wide melting peak, due to the presence of ethylene units randomly distributed along the macromolecular chain deriving from propylene.

With PPR, crystalline content increases over time but analyses of the surface are influenced by the presence of a powder formed during aging, which have lower melting and crystallization temperatures than PPR (see Appendix A). As it was mechanically removed, analyses of the surface of the pipe after its removal might anyway be influenced by the presence of residues. 

As in the case of PERT, ΔH_c_ is always higher than ΔH_m_ and is less influenced by aging, even if also, in this case, PPR becomes more crystalline over time. 

The increase in crystallinity observed is in good agreement with microscope observations: higher crystallinity implies that materials become stiffer and more fragile, leading to the formation of cracks.

### 3.3. Impedance Analyses

#### 3.3.1. PERT Multilayer Samples

The results of the impedance analyses of PERT samples are shown in Table 3.

As PERT has been studied only with the IS technique (due to the high electrical resistance of the material), the equivalent circuit that better fits the experimental data is an RC_dl_ series, whose only C_dl_ variations are predictive of the degradation of the material (Table 3).

The test was repeated in triplicate, evidencing that aged samples always show a higher double-layer capacitance, connected with the inner plastic surface modification/corrosion (with yields to an increase in the real area) due to the degradation process carried out by ClO_2_. Chlorine dioxide probably reacts with polymer and antioxidant compounds, which damages and removes part of the material, increasing inner surface roughness.

#### 3.3.2. Copper Samples

Before investigating the Cu pipe inner surface, Cyclic Voltammetry (CVs) have been used to identify the potential at which it is possible to highlight the surface degradation process that, for our purpose, is directly connected with the modification, both, of the surface area, affecting *C*_dl_, and the electron transfer process for Cu oxidation, acting on R_ct_. Figure 8 shows CVs for two different samples: new pipe (black continuous curve) and aged 8 weeks (dotted curve) Cu tubes. It emerges that aged Cu shows higher values of current density, both for oxidation and reduction processes, probably connected with a more electro-active metallic surface after the contact with ClO_2_. In particular, at the start potential of 0 V, the current is almost zero for both samples; then, during the anodic scan, current increases much more for aged than pristine samples, due to the corrosion process occurring at the Cu surface; during the backward scan, it is possible to see a cathodic bump at about 0 V, probably due to the reduction of Cu ions formed during the anodic scan. After this analysis, the selected investigating potentials for EIS measurements have been 0, 0.05V, and 0.1 V vs. SCE.

Figure 9a,b shows the Nyquist diagram for new (continuous curve) and aged (dotted curve) samples at different polarization potentials for the frequency range selected to highlight the behavior of electron transfer resistance, R_ct_, and double layer capacitance, C_dl_. Fitting was carried out using an R_1_ in series to a parallel R_ct_CPE. The use of a constant phase element (CPE) is necessary due to the presence of rough metallic surfaces and pores, and in the case of the aged sample, due to the corrosion product film on the metallic surface, thus evidencing a diffusion step in the same frequency domain.

The reduction in the R_ct_ values (from the semicircle amplitude) is evident, increasing the polarization potential from 0 to 0.05 V versus SCE (Figure 9a vs. Figure 9b), due to the increase of the Cu corrosion process rate. At 0.1 V, the Nyquist plot presents two almost overlapped semicircles, as at this higher potential, the t = 0 sample is prone to the anodic degradation process. At a lower potential, this phenomenon is much more evident for the aged Cu (see R_ct_ in Table 4), as its surface is already modified and damaged by ClO_2_ action, and at 0.05 V vs. SCE, the anodic polarization accelerates the corrosion process. 

The higher R_ct_ value for Cu at t = 8 weeks compared to t = 0, when polarized at 0.05V, is probably connected with the CPE exponent value, which is about 0.6 for Cu at t = 0 at all potentials and about 0.5 for Cu at 8 weeks of aging at 0 and 0.1 V, while it becomes 0.23 at 0.05 V (see 4th column Table 4). This is most likely due to the modification of pore sizes and the shape present in the corrosion film on the Cu surface; a phenomena that can deeply affect the diffusion of ions back and forth towards the underlying metallic surface, thus reducing the electron transfer rate evidenced by the increase of R_ct_. In any case, this behavior is strictly connected with the film layer on the Cu surface produced during the aging process in the presence of ClO_2_. No such behavior has been detected for the pristine Cu. Lastly, the C_dl_ values for the pristine sample are smaller, of about one order of magnitude, than the aged values (Table 4), thus indicating a more corrugated metallic surface due to the damaging action of ClO_2_.

## 4. Conclusions

Both of the results from the metallic and plastic pipes show that pipes treated with ClO_2_, even after a relatively short aging period in relatively mild conditions (70 °C—1 mg/L ClO_2_) in a semi-closed system, show strong surface degradation. The physical degradation is accompanied by changes in the chemical structure of plastics (PERT and PPR) and in the chemical composition of the surface of metal pipes (copper and galvanized steel). Metals are also visible on the surface of plastic pipes, due to transportation in the water flux. DSC analyses on plastic pipes show an increase in crystallinity, leading to increased brittleness, whereas impedance measurements particularly evidenced by the increased values of *C*_dl_, are directly connected with an increase of inner surface roughness. In the case of aged metal tubes, the electron transfer rate for the oxidation process is accelerated, due to the embrittlement and damaging of metal surfaces treated with ClO_2_ disinfectant agent. The results obtained suggest that issues related to progressive deterioration of pipes caused by ClO_2_ should be seriously taken into account, especially considering that the products of degradation, both in terms of microplastics and of metallic species, are likely to reach common drinking water in houses.

## Figures and Tables

**Figure 1 ijerph-16-04582-f001:**
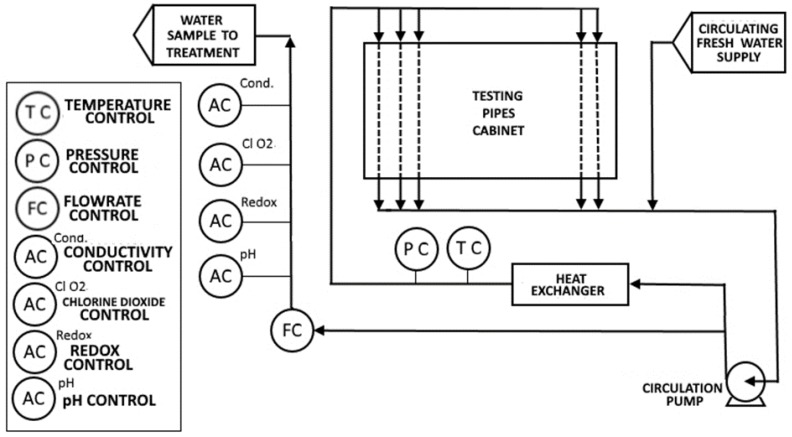
Schematic representation of the testing apparatus.

**Figure 2 ijerph-16-04582-f002:**
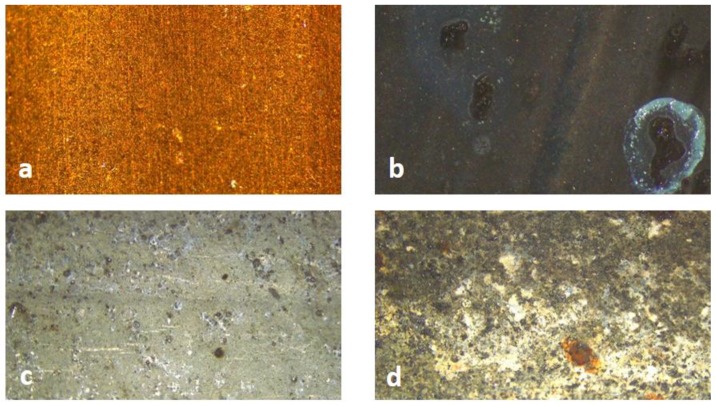
Cu pipe—new (**a**), Cu pipe—8 weeks (**b**), galvanized steel—new (**c**), and galvanized steel—8 weeks (**d**).

**Figure 3 ijerph-16-04582-f003:**
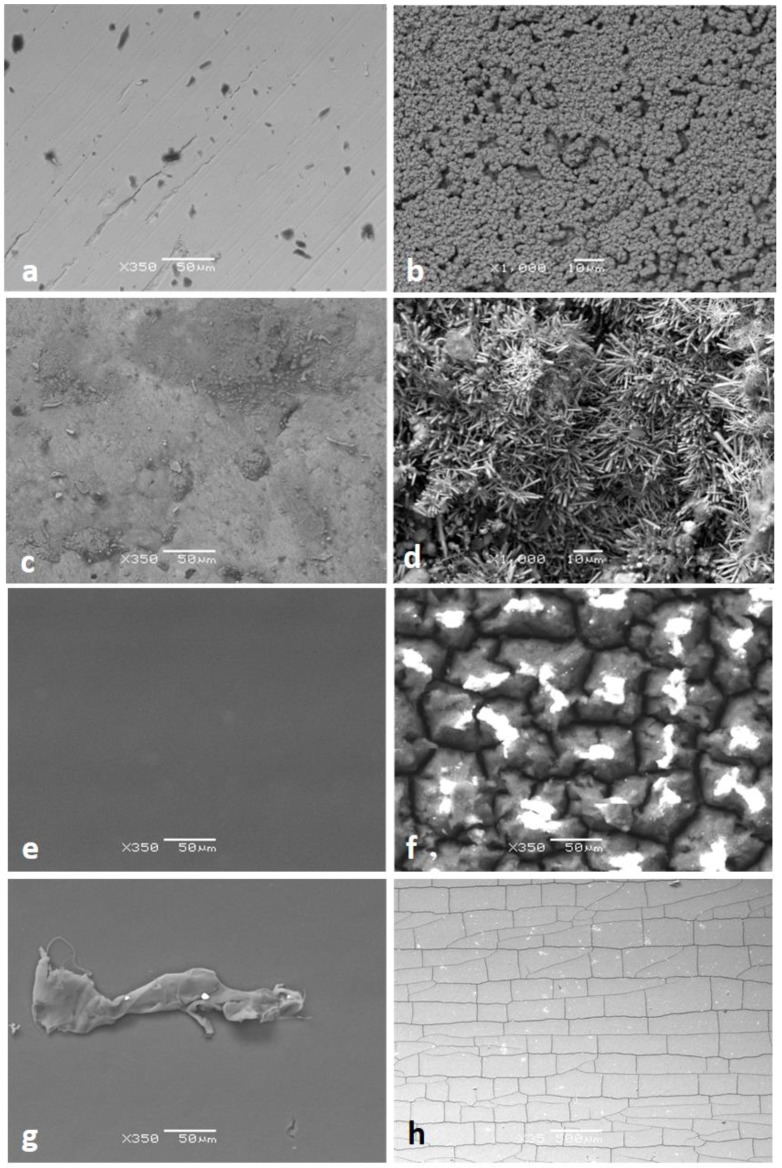
Cu pipe—new (350X) (**a**), Cu pipe—8 weeks (1000X) (**b**), galvanized steel—new (350X) (**c**), galvanized steel—8 weeks (1000X) (**d**), random polypropylene (PPR)—new (350X) (**e**), PPR—8 weeks (350X) (**f**), polyethylene of raised temperature (PERT)—new (350X) (**g**), and PERT—8 weeks (350X) (**h**).

**Figure 4 ijerph-16-04582-f004:**
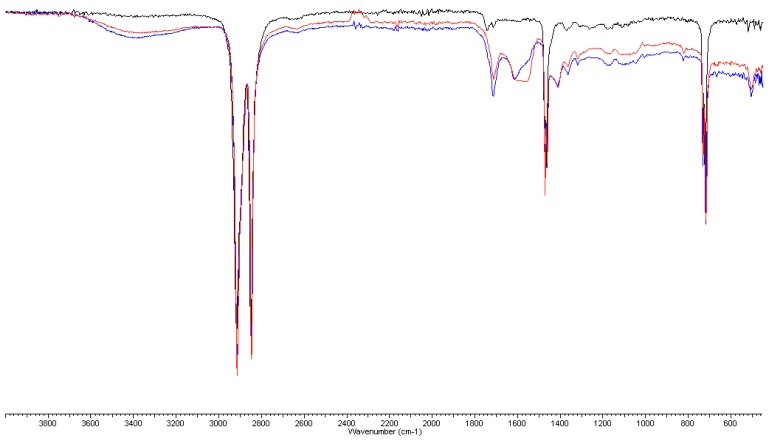
FT-IR spectra of PERT inner surface: new (**black**), 4 weeks (**red**), and 8 weeks (**blue**).

**Figure 5 ijerph-16-04582-f005:**
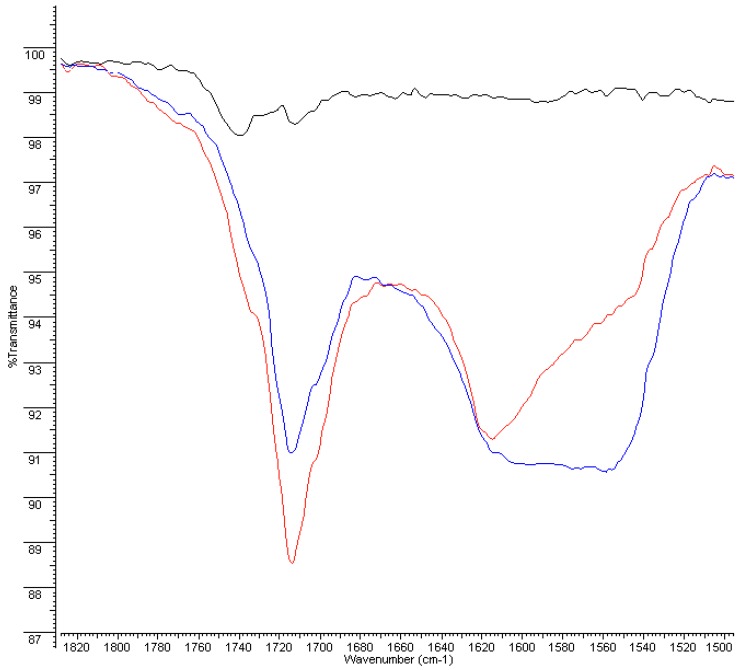
FT-IR spectra of PERT inner surface: new (**black**), 4 weeks (**red**), and 8 weeks (**blue**)—magnification of 1830–1500 cm^−1^ region.

**Figure 6 ijerph-16-04582-f006:**
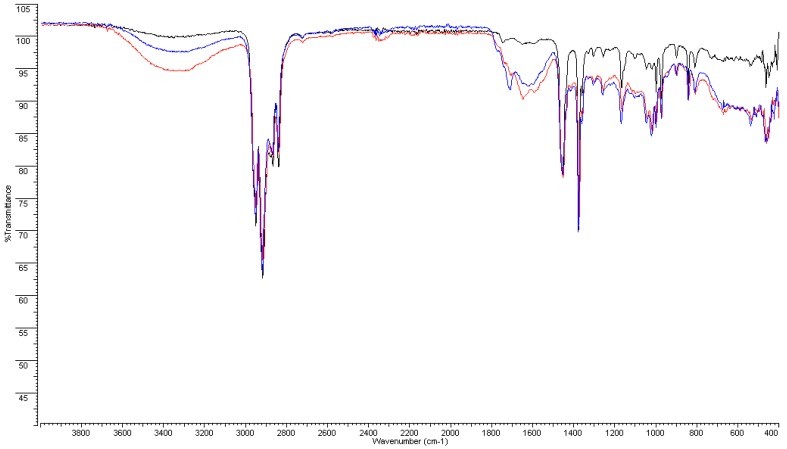
FT-IR spectra of PPR inner surface: new (**black**), 4 weeks (**red**), and 8 weeks (**blue**).

**Figure 7 ijerph-16-04582-f007:**
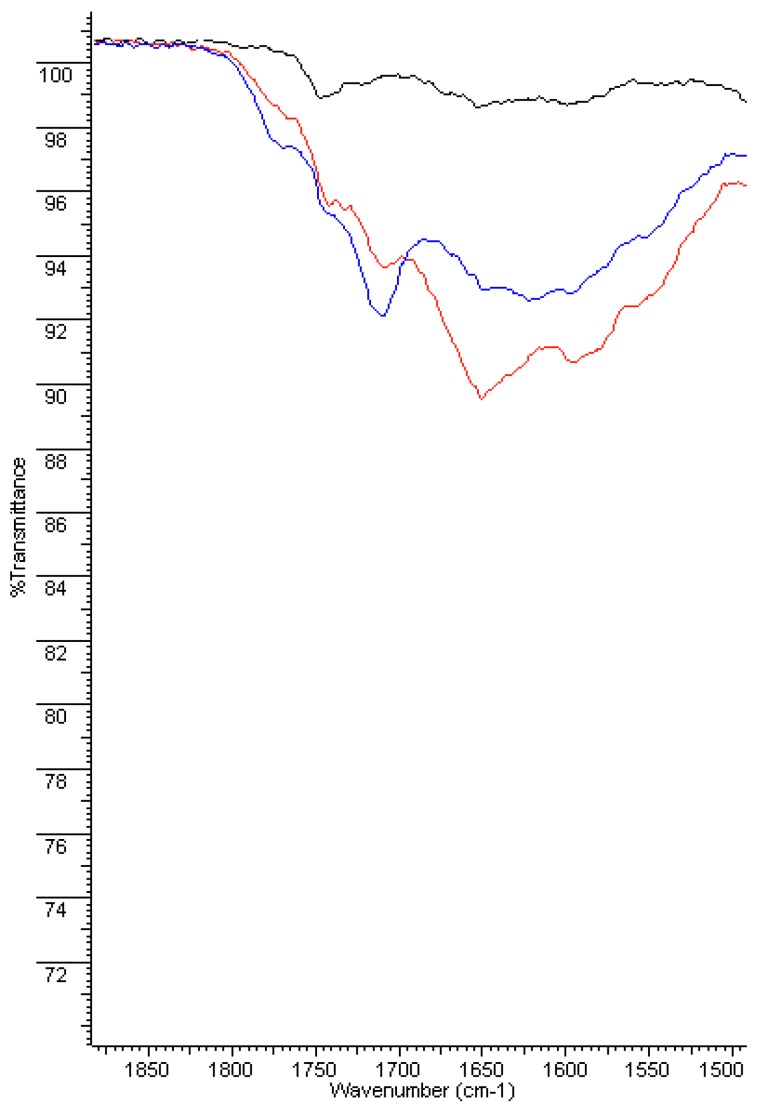
FT-IR spectra of PPR inner surface: new (**black**), 4 weeks (**red**), and 8 weeks (**blue**)—magnification of 1900–1500 cm^−1^ region.

**Figure 8 ijerph-16-04582-f008:**
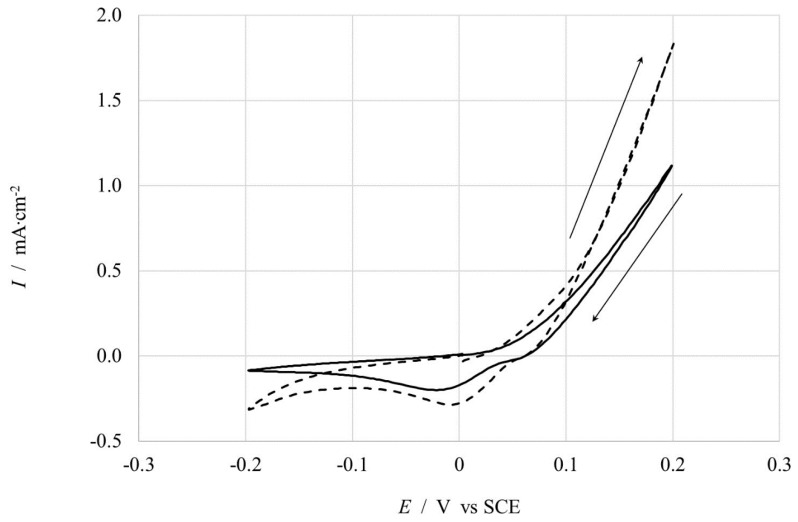
CV in 0.5 M KNO_3_ at 20 mV s^−1^; new pipe (**continuous**), Cu aged 8 weeks (**dotted**).

**Figure 9 ijerph-16-04582-f009:**
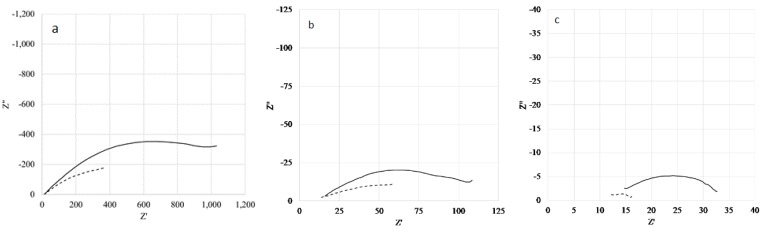
Cu new pipe (continuous curve) and after 8 weeks aging (dotted curve) at 0.0 V (**a**), 0.05 V (**b**), and 0.1 V (**c**) vs. the saturated calomel electrode (SCE).

**Table 1 ijerph-16-04582-t001:** Energy dispersive spectrometer (EDS) analyses on pipes.

Material	Elements—New Pipe (*w*/*w*)	Elements—Pipe after 8 Weeks (*w*/*w*)
Copper	Cu (100%)	Cu, O
Galvanized Steel	Zn (97.7%) Fe (2.3%)	Zn (60.3%), O (21.9%), Fe (13.6%), Cu (2.4%), Cl (0.8%), Al (0.7%), K (0.3%)
PPR	C, O (trace)	O (71.3%), C (26.2%), Ti (0.9%), Fe (0.4%), Si (0.3%), Cu (0.3%), Cl (0.1%)
PE-RT	C, O (trace)	O (72.1%), C (27.0%), Cu (0.5%), Cl (0.1%), Si (0.1%)

**Table 2 ijerph-16-04582-t002:** Thermal data obtained via differential scanning calorimetry (DSC) on PERT and PPR pipes.

Sample	Aging (Weeks)	Position	*T* _m_	Δ*H*_m_	*T* _c_	Δ*H*_c_	χ_c_
			°C	J/g	°C	J/g	%
PERT	0		127.9	−133.4	115.8	155.0	46.5
	4	Surface	126.8	−143.8	116.6	164.5	50.1
		Bulk	127.7	−155.6	116.3	159.7	54.3
	8	Surface	127.1	−155.1	116.9	168.0	54.1
		Bulk	127.2	−159.0	116.4	168.1	55.4
PPR	0		141.5	−50.9	111.2	60.7	24.6
	4	Surface	143.6	−58.5	111.6	62.8	28.3
		Bulk	143.4	−54.2	111.0	64.2	26.2
	8	Surface	145.9	−45.0	111.0	63.4	21.7
		Bulk	141.7	−62.7	111.8	63.7	30.3

**Table 3 ijerph-16-04582-t003:** C_dl_ values for the different tested samples: new pipe and aged for 8 weeks.

Sample	*C*_dl_, F
PERT new pipe	(3.00 ± 0.10) 10^−11^
PERT aged 8 weeks	(4.40 ± 0.04) 10^−11^

**Table 4 ijerph-16-04582-t004:** R_ct_ for Cu of new pipes and of pipes aged 8 weeks.

Cu	*R*_ct_, Ω	*CPE*, F	α
New pipe 0 V	(1486 ± 19)	(4.35 ± 0.05) 10^−4^	(0.578 ± 0.002)
Aged 8 weeks 0 V	(1279 ± 127)	(2.30 ± 0.06) 10^−3^	(0.453 ± 0.006)
New pipe 0.05 V	(108.9 ± 0.8)	(1.37 ± 0.03) 10^−3^	(0.450 ± 0.002)
Aged 8 weeks 0.05 V	(121 ± 6)	(1.18 ± 0.03) 10^−2^	(0.227 ± 0.005)
New pipe 0.1 V	(22.6 ± 0.3)	(6.6 ± 0.3) 10^−4^	(0.524 ± 0.008)
Aged 8 weeks 0.1 V	(6.7 ± 0.5)	(3.6 ± 0.6) 10^−3^	(0.423 ± 0.003)

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
