# Peer review of "Chlorine Dioxide Degradation Issues on Metal and Plastic Water Pipes Tested in Parallel in a Semi-Closed System"

_ijerph, 2019, doi:10.3390/ijerph16224582_

Round 1
Reviewer 1 Report
Comments
Lines 23-25 Results show that ClO2 has a deep effect on all the materials tested (plastics and metals) and that severe damage occurs due to its strong oxidizing power...Base deep effect and damage, which and how much?
Lines 106-108 High quantities of disinfectant?, in relatively short?...How much?
Line 119 The pipes were tested for 8 weeks...Basis choice of number of weeks
Line 212 In particular, the black color of Cu pipes seems to indicate the formation of Cupric oxide...Indicates or seems to indicate?
The text of the paper must include a more detailed description of the DSC and FT-IR methods.
Point out conclusions
Lines 372- 375 must be removed from the item conclusions.
It is a very interesting work since it poses a public health risk, which is not usually addressed
Author Response
The authors wish to thank the reviewer, first of all for appreciating the paper and also for the useful and interesting comments, that will help improving the quality of the paper itself.
In relation to the comments
Lines 23-25: we added an explanation on what happens on metals and plastics
Lines 106-108: the sentence refers to citations 26 and 27. In citation 26, authors used water containing 7-400-4000 ppm of chlorine to study PE degradation for 5 weeks in the case of 4000 ppm and for 15 weeks when using 400 ppm.
In citation 27, PERT was tested using water containing 0-1-25-100 ppm of chlorine for 270 days at 70°C and observe a rapid depletion (after 70 days) of the stabilizing system of PERT when using 25-100 ppm of chlorine. We did not add all these data in the paper in order not to make it too long
Line 119: the reason for choosing 8 weeks of testing is due to experimental observation during the test. We tested 0-1-2-3-4-6-8 weeks and stopped after 8 weeks because preliminarly observation (optical microscope and SEM) showed that the inner surface of the pipes had been severely damaged
Line 212: the referee is right, now "indicates" is used
We added details regarding DSC and FT-IR methods
Reviewer 2 Report
In this paper, the pipeline corrosion caused by chlorine dioxide as disinfectant was investigated in detail, and the conclusions provided new data and theoretical support for the choice of disinfectant in practical engineering application. But there are still some areas that need further improvement.
The results of corrosion are characterized and described in detail in the paper, but in the discussion part, there is a lack, especially in the discussion of the causes and mechanism of corrosion. It is suggested to add. In this paper, if the common disinfectant chlorine can be compared synchronously through the test, it will be better. Under the current situation, whether the existing literature can be cited for comparison, which is conducive to the evaluation of the corrosion degree of pipelines by chlorine dioxide. Is there any established standard for pipeline corrosion? There is no description about water quality change by pipeline corrosion. I don't know whether relevant research has been carried out. If possible, it is suggested to add. The language needs further polish.Author Response
Authors wish to thank the reviewer for the interest shown and the comments. Here are the answer to the points raised
1) we detailed the mechanism proposed in previous literature for polyolefins failure in presence of chlorine dioxide in the final part of FT-IR section. Regarding metals, oxidation is the responsible for the corrosion and, unfortunately, we did not find significant research on this topic. Given the aims of this specific paper, in the future we will try to deepen the study both on plastics and on metals separating the two "families" of materials to better focus on each
2) The referee is right, since a comparison between different disinfectans could be very useful. This paper was dedicated to chlorine dioxide, that is less studied than chlorine: after the end of the tests, we started studying also other disinfectants such as chlorine and monochloramine.
3) Standards, unfortunately, are not "all around": some of them, such as F2023 (cited in the paper) and F2263 use only "time to failure" as an indicator of pipes performance. Others, such as ASTM D2513 determine 8 levels of cracking for plastics but do not deal with metals and do not deal with the chemicals formed due to degradation. OIT (ISO 11357-6 or ASTM D3895) only indicate a time for oxidation of the material, not directly correlated to corrosion.
4) After obtaining such results, we also decided to consider studying water quality: in January 2020 we will start using filters where water leaves the testing chamber to collect microplastics and we will check metals to deepen this issue, that is fundamental